# Self-cleaning and surface chemical reactions during hafnium dioxide atomic layer deposition on indium arsenide

Rainer Timm [1], Ashley R. Head[1,3], Sofie Yngman[1], Johan V. Knutsson[1], Martin Hjort[1,4], Sarah R. McKibbin[1], Andrea Troian[1], Olof Persson[1], Samuli Urpelainen[2], Jan Knudsen[1,2], Joachim Schnadt[1] & Anders Mikkelsen[1]

Atomic layer deposition (ALD) enables the ultrathin high-quality oxide layers that are central to all modern metal-oxide-semiconductor circuits. Crucial to achieving superior device performance are the chemical reactions during the first deposition cycle, which could ultimately result in atomic-scale perfection of the semiconductor–oxide interface. Here, we directly observe the chemical reactions at the surface during the first cycle of hafnium dioxide deposition on indium arsenide under realistic synthesis conditions using photoelectron spectroscopy. We find that the widely used ligand exchange model of the ALD process for the removal of native oxide on the semiconductor and the simultaneous formation of the first hafnium dioxide layer must be significantly revised. Our study provides substantial evidence that the efficiency of the self-cleaning process and the quality of the resulting semiconductor–oxide interface can be controlled by the molecular adsorption process of the ALD precursors, rather than the subsequent oxide formation.

[1] Division of Synchrotron Radiation Research, Department of Physics, and NanoLund, Lund University, Box 118, 221 00 Lund, Sweden. [2] MAX IV Laboratory, Lund University, Box 118, 221 00 Lund, Sweden. [3] Present address: Chemical Sciences Division Lawrence Berkeley National Laboratory, Berkeley, CA 94720, USA. [4] Present address: Division of Solid State Physics, Department of Physics, and NanoLund, Lund University, Box 118, 221 00 Lund, Sweden. Correspondence and requests for materials should be addressed to R.T. (email: rainer.timm@sljus.lu.se)

II–V semiconductors such as indium arsenide (InAs) or gallium arsenide (GaAs) are central candidates for next-generation high-speed/low-power electronics, mainly because their electron mobility is much larger than that of Si[1–4]. However, one key challenge in using these materials is the detrimental effect of high interface trap densities, which are caused by unwanted oxides, atomic vacancies, dimers, or other atomic-scale interface defects[5–7] formed during device fabrication. These can in particular limit the performance of nanoscale metal-oxide-semiconductor field effect transistors (MOSFETs), which critically relies on the structural quality of the semiconductor–oxide interface[1,2]. Strong improvement on this issue has been obtained by depositing ultra-thin barriers of hafnium dioxide (HfO$_2$)[8,9] which not only has a high dielectric constant (high-$k$), but also potentially can substitute the native oxide with a well-controlled, stable, and defect-free semiconductor–oxide interface.

The standard approach for depositing high-$k$ oxide layers on semiconductors is based on atomic layer deposition (ALD). The ALD procedure consists of alternating self-limiting half-cycles that use two different precursors[10–13]. HfO$_2$ can be deposited using ALD by first exposing the substrate to a pulse of tetrakisdimethylamido-hafnium (TDMA-Hf), then to a pulse of water, and then repeating this alternating process to produce homogeneous high-$k$ oxide films of atomic monolayer precise thickness. During this ALD process, unwanted native oxides are near-completely removed via the so-called self-cleaning effect[9,14–20]. X-ray photoelectron spectroscopy (XPS) has been proven as a uniquely important tool for investigating the chemical composition of the substrate surface and the semiconductor–oxide interface[7,15–19] and XPS studies have also shown that the self-cleaning effect occurs within the first ALD half-cycle[7,21,22]. Complementary information especially about the nature of organic species at the surface has been obtained by Fourier transform infrared spectroscopy (FTIR)[20,23–25]. However, until now experimental studies have been limited to obtain information about the surface chemistry before and after an ALD half-cycle, but not during the deposition itself. The reason for this limitation is mainly the millisecond timescales and the millibar pressure range of typical ALD setups, which are not amenable to either conventional XPS instrumentation that requires ultra-high vacuum (UHV) conditions or FTIR, which has longer measurement times and will be dominated by the gas in the chamber. This lack of knowledge about the surface chemical reactions that occur during the first cycle of the ALD process[26] is especially unfortunate, because the first cycle is central in determining the properties of the interface, as shown by a number of studies[3,20–23,27] and the interface quality between III–V semiconductors and high-$k$ oxides has still not reached a satisfactory level for real device applications.

Here, we present a time-resolved study of the self-cleaning effect occurring during the first few ALD half-cycles of HfO$_2$ deposition on InAs using ambient-pressure XPS (AP-XPS)[28–31] with an in-vacuum gas cell design[29]. In this setup we can monitor chemical reactions between a solid surface and reactive gases flowing through the system under realistic synthesis conditions. We find that the formation of HfO$_2$ on the InAs surface during the first ALD half-cycle is preceded by the incorporation of hafnium (Hf) atoms in a different chemical configuration, which we interpret as molecular adsorption of TDMA-Hf. These findings challenge the established view of the ALD process. Current understanding of how the high-$k$ oxide layer forms by ALD is based on the ligand exchange model, where the metalorganic precursor molecule splits off one of its ligands and instead forms a bond to an oxygen (O) atom from the sample surface, thereby replacing the unwanted native oxide with a high-quality high-$k$ oxide film[10,11,13,32]. In contrast, our results show that the molecular adsorption of the Hf-precursor molecules occurs prior to the ligand split-off, and that the native oxide is desorbed already during this precursor adsorption step, prior to Hf–O bond formation. In fact, we observe an almost complete removal of the native oxide, including several As-oxide and In-oxide components. Our results have three important implications: (1) They call for significant revisions of the established ligand exchange model to describe the ALD process. (2) They yield an understanding of how a typically several monolayer thick native oxide layer is removed while only a sub-monolayer thick high-$k$ oxide film is formed—this has been one of the main shortcomings of the ligand exchange model. (3) They point to new strategies for improving the interface quality between III–V semiconductors and high-$k$ oxides by optimizing the ALD kinetics. While we chose to study ALD of HfO$_2$ on InAs because of its high technological relevance, our experimental approach using AP-XPS and the reported mechanistic insights on semiconductor–oxide interface formation are not limited to this system, but are relevant for understanding surface chemistry processes during ALD in general.

## Results

**Arsenic-oxide removal during the first ALD half-cycle**. As the removal of the native oxide on the InAs surface is a central feature of the ALD process, we begin by monitoring this process during the first deposition of TDMA-Hf and follow the time evolution of the arsenic (As) 3d core level spectra, which has a clear signature of both As bound to oxygen and As bound in the InAs crystal. The X-ray photoelectron (XP) spectra reveal a complete removal of the As-oxide component (Fig. 1a) within a timeframe of about 20 s (Fig. 1b). In commercial ALD reactors, the reaction time is typically shorter and also the onset of the reaction is more abrupt, which would make XPS observations of the reaction process difficult. In our setup we obtain a slower and delayed reaction because of adsorption of the precursor on the walls of the gas pipes leading to the reaction cell. This is due to the low vapor pressure of the TDMA-Hf precursor, and because we keep the gas pipes unheated, as discussed in detail in Supplementary Fig. 1 and Supplementary Note 1.

To verify that the chemical reactions which occur in our in situ synchrotron ALD setup are the same as those in standard ALD machines, we acquired UHV XP spectra at high resolution following each ALD half-cycle (see Supplementary Fig. 2 and Supplementary Note 2). We thereby obtained detailed spectra of relevant core levels measured under static sample conditions. The spectra of As 3d and indium (In) 3d core levels before and after the initial TDMA-Hf deposition show a complete removal of the native As$^{3+}$- and As$^{5+}$-oxide components as well as a strong reduction of the amount of In oxide (Supplementary Fig. 2). These results, and the shape of the obtained spectra, are in good agreement with previous XPS studies obtained ex situ on HfO$_2$ thin films deposited on InAs in commercial ALD setups[15,33,34] From this we can conclude that, while the ALD process is slower in our AP-XPS setup to allow a time-resolved observation, the interface structure and composition of the deposited thin films and therewith the products of the involved surface chemistry are fully comparable to those produced with conventional ALD equipment.

**Temporal evolution of the hafnium surface chemistry**. To reveal the different stages of Hf surface chemistry during the HfO$_2$ formation, we analyze the temporal evolution of the In 4d and Hf 4f core levels during ALD deposition (Fig. 2), with a special focus on the shift in Hf binding energy. During the first TDMA-Hf deposition, a strong Hf signal appears. The duration and delay of this reaction are comparable to that of the As-oxide

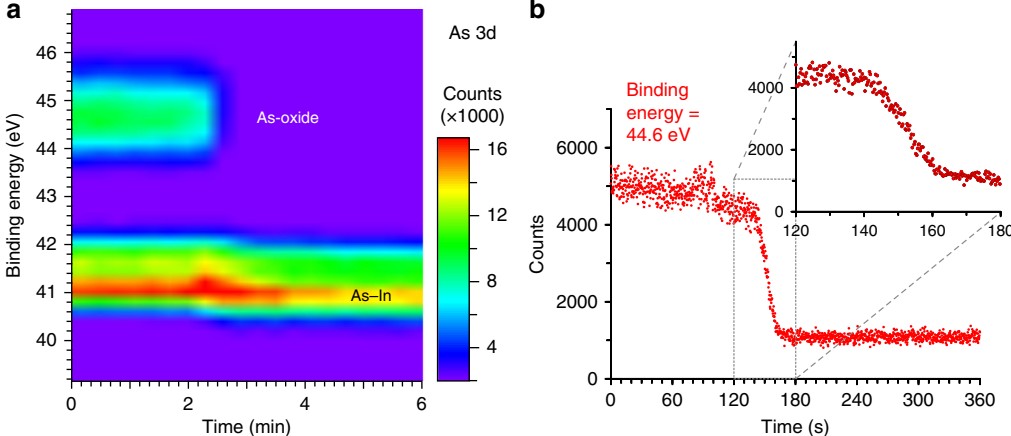

**Fig. 1** Time-resolved arsenic-oxide removal. **a** Time evolution of As 3d XPS core level spectra during exposure to TDMA-Hf. Components corresponding to As oxide and to As bound to In (bulk) can clearly be distinguished at binding energies of 44.6 eV and 41 eV, respectively. The oxide component is removed completely after an initial delay of about 2 min, accompanied by a shift in binding energy of the bulk component. The time interval between subsequent spectra was 17 s. **b** Time evolution of the XPS peak intensity at a binding energy of 44.6 eV, corresponding to the maximum of the As-oxide peak, during exposure to TDMA-Hf. The inset shows a zoom-in at the time period of strong oxide removal. The timescales correspond to the opening of the gas inlet valve. Spectra were obtained at a photon energy of 320 eV, a gas pressure of about 1 Pa, and a sample temperature of 200 °C (**a**) and 220 °C (**b**)

removal (see Fig. 1, Supplementary Fig. 1, and Supplementary Note 1). Upon subsequent ALD half-cycles, the Hf components shift to lower binding energies during water deposition (Fig. 2b) to higher binding energies during TDMA-Hf deposition (Fig. 2c). These shifts correspond to two stable Hf 4f configurations (see Supplementary Fig. 3 and Supplementary Note 3 for details). A similar peak shift trend is observed for the O 1s AP-XPS peak (Supplementary Fig. 4a, b). Shifts in binding energy are often explained as chemical shifts, due to, e.g., a change in the oxidation state of the corresponding species[7,15,18]. However, because both the Hf 4f and O 1s peaks shift to lower binding energies during water deposition (and back during the following half-cycle), these shifts cannot be explained by the occurrence of redox reactions between Hf and O atoms (as that should induce shifts in opposite directions). We instead attribute the shifts to an increasing occurrence of hydroxide groups, which results in a less ionic character of the Hf–O bonds[35]. Indeed, the O 1s spectra show a significant O–H component after water deposition, which disappears again upon subsequent TDMA-Hf deposition (Supplementary Fig. 4c, d, see also Supplementary Note 4). Although such a hydroxylation has been postulated in standard ALD models[11,36], direct experimental verification has been missing until this study.

We observe a signature that indicates the presence of a hitherto unreported, and unexpected intermediate step in the ALD reaction, which manifests itself as a significant energy shift of the Hf core level peak at the onset of Hf incorporation in the sample surface (Fig. 2a, d). When the first Hf species bond to the surface, as indicated by the increase in peak size (orange, yellow and green curves in Fig. 2d), the observed binding energy decreases during three spectra (taken within 32 s), and then increases again, remaining stable thereafter (dark purple curves). It is important to point out that this observed shift in binding energy cannot be explained by a possible chemical shift of the overlapping In 4d peak, as the In-related signal is too weak and the peak shape is dominated by the Hf-related doublet. This is shown in Supplementary Fig. 5 and discussed in detail in Supplementary Note 5. While the final configuration of the Hf 4f peak can be attributed to Hf–O bonds from the HfO2 film, the initially observed Hf atoms with a ~0.3 eV smaller binding energy must be in a different chemical state, which is the precursor molecule configuration. Thus, the ALD reaction is found to

proceed in two distinct steps: (1) adsorption of Hf-containing molecules to the InAs substrate and (2) subsequent formation of Hf–O bonds during the formation of the HfO2 layer. It can also be observed that the intensity of the XP spectra derived from the first species is already two thirds of the intensity of the final Hf peak (Fig. 2d). This emphasizes the importance of a significant amount of adsorbed Hf molecules in order to start the reaction toward a Hf–O layer. During later TDMA-Hf deposition half-cycles, no such adsorption step is observed. This type of two-step ALD mechanism within the first half-cycle has never been resolved or reported before and indicates that the commonly used single step half-cycle model of ALD[10,11] must be revised, as we discuss below.

**Correlation of Hf surface chemistry with In- and As-oxide removal and Fermi-level unpinning**. We are now ready to investigate the oxide removal in more detail and to correlate it with the temporal evolution of the Hf surface chemistry, by correlating As 3d, In 3d, and In 4d / Hf 4f core level spectra that are obtained alternately during the first deposition of TDMA-Hf (Fig. 3). The removal of the native oxide is found not to occur uniformly, but in successive self-cleaning steps for different oxide components: two As-oxide components can be distinguished in the As 3d spectra (Fig. 3a), while only one broad oxide peak can be fitted in the In 3d spectra (Fig. 3c), likely consisting of several sub-oxides with overlapping peaks[15]. The As5+ component (as in As2O5) decreases first and is almost completely removed when the removal of the As3+ component (as in As2O3) sets in, while In oxides are removed in between (Fig. 3d).

The onset of the native oxide removal, as monitored in As 3d spectra (Fig. 3a, curves 05 and 07), occurs at the same time as the first adsorption of Hf species on the InAs surface, indicated by the increase in Hf 4f peak size and the shift to lower binding energies (Fig. 3b, curves 06 and 08). The most significant oxide removal is observed at the same time as the strongest signal from adsorbed Hf molecules is obtained (Fig. 3b, curve 08, showing the lowest binding energy of the Hf 4f peak). No As oxide is found on the surface anymore (Fig. 3a, curve 11) when the Hf 4f peak shifts to higher binding energies, indicating the formation of a HfO2 layer (Fig. 3b, curves 12 and 14). Thus, the Hf–O bond formation does not occur simultaneously, but instead after the native oxide

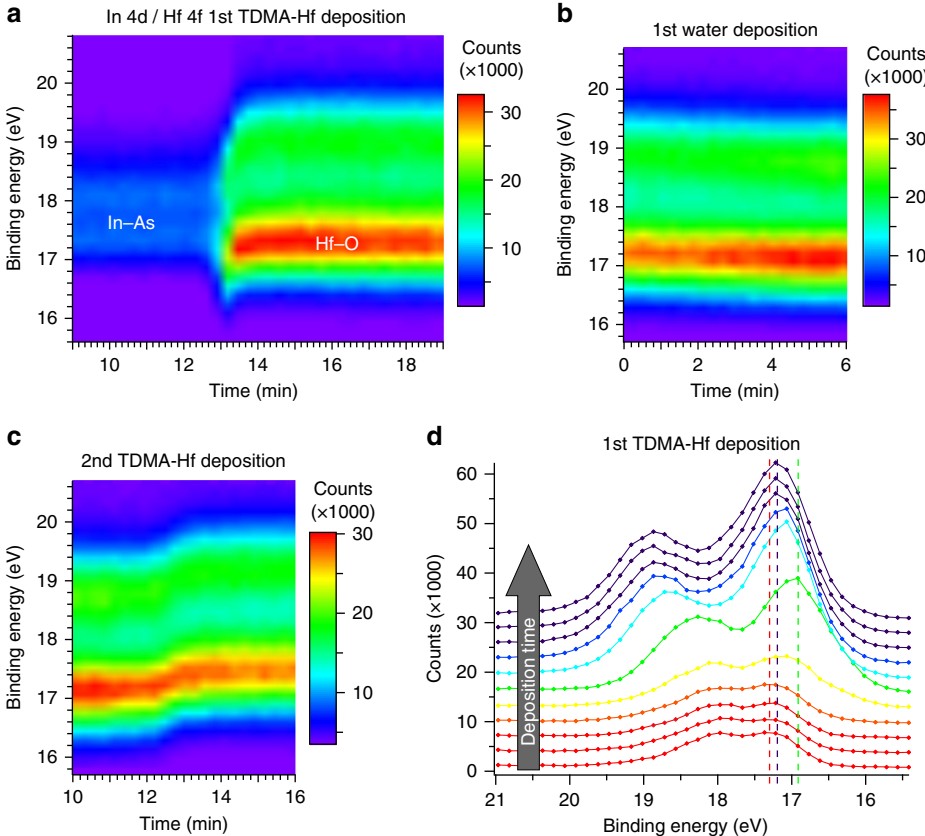

**Fig. 2** Time-resolved Hf surface chemistry. Time evolution of Hf 4f and In 4d core level XP spectra, which are overlapping in binding energy, during the initial half-cycles of the ALD reaction, i.e., first deposition of TDMA-Hf (**a**), first deposition of water (**b**), and second deposition of TDMA-Hf (**c**). During the first half-cycle (**a**), first only a weak signal corresponding to In–As and In-oxide components can be seen, until the reaction sets in. Afterward, the spectra are dominated by the Hf signal, because the photoionization cross-section for Hf 4f is seven times larger than that for In 4d (at the photon energy of 300 eV used here)[37]. At the onset of the Hf 4f peak, a significantly lower binding energy is observed only for a few seconds during the initial phase of the reaction. **d** Subsequent In 4d/Hf 4f XP spectra of the time evolution series shown in **a** shortly before (red), during (orange, yellow, green, cyan, blue), and after (dark purple) this initial phase of Hf incorporation. Peak binding energies of some distinct spectra are indicated by dashed lines (in corresponding colors). The time delay between individual spectra was 16 s. The timescales correspond to the opening of the gas inlet valve. Spectra were obtained at a photon energy of 300 eV, a gas pressure of about 0.3 Pa, and a sample temperature of 220 °C

removal, and the self-cleaning effect during the first ALD half-cycle has to be attributed to the Hf precursor molecules that are adsorbed on the surface in the first step of the reaction.

A very important question is how the removal of the native oxide and the introduction of the HfO₂ layer influence the position of the Fermi level relative to the InAs bandgap. The native oxide typically gives the InAs surface an n-type character due to Fermi-level pinning[38] which is undesirable for any attempts of controlling device doping levels and thus electrical performance. The position of the Fermi level relative to the valence band edge at the sample surface can be obtained from the binding energies of the InAs-related peaks in the As 3d (Fig. 3a) and In 3d (Fig. 3c) core level spectra[39]. Although the accuracy of absolute binding energies is limited by the experimental energy resolution and also affected by the fitting procedure (e.g., the assumed shape of the background), the As 3d and In 3d spectra all show the same three trends for the relative shifts in energy (Fig. 3d): (1) The measured binding energies after the ALD reaction are smaller than those of the starting substrate, which implies that the sample surface has become less n-type. However, the observed shift of about 0.05 eV is rather small compared to the InAs bandgap of 0.35 eV, indicating that the Fermi level at the InAs surface remains partially pinned even after native oxide removal. (2) The binding energies are found to first increase slightly during the initial reaction, i.e., during the removal of the

native oxide and the adsorption of Hf molecules. (3) The main decrease of the binding energies happens directly after the oxide removal has finished, i.e., while the Hf–O bond formation occurs. The latter two observations indicate that the energy position of the InAs surface relative to the Fermi level is not primarily determined by the amount of native oxide, but rather by the Hf surface chemistry. The efficiency of the self-cleaning effect and the position of the Fermi level at the interface between the high-*k* oxide and the semiconductor are crucial parameters for the performance of III–V semiconductor MOSFETs. Both parameters are found to be controlled by the Hf precursor molecular adsorption, which has been observed here for the first time.

**Temperature dependence of the self-cleaning effect.** As the sample temperature is an important ALD parameter, both regarding film quality and feasibility of the ALD process, we lastly explore the effects of a sample temperature that is too low for native oxide removal and HfO₂ layer formation. In contrast to the very efficient self-cleaning effect discussed above, we observe that decreasing the sample temperature to below 200 °C leads to only a partial removal of the native oxide (Fig. 4a). In addition, the intensity of the InAs As 3d peak decreases significantly during the TDMA-Hf deposition, as a result of the attenuation of the InAs signal by a deposited surface layer of more than one monolayer

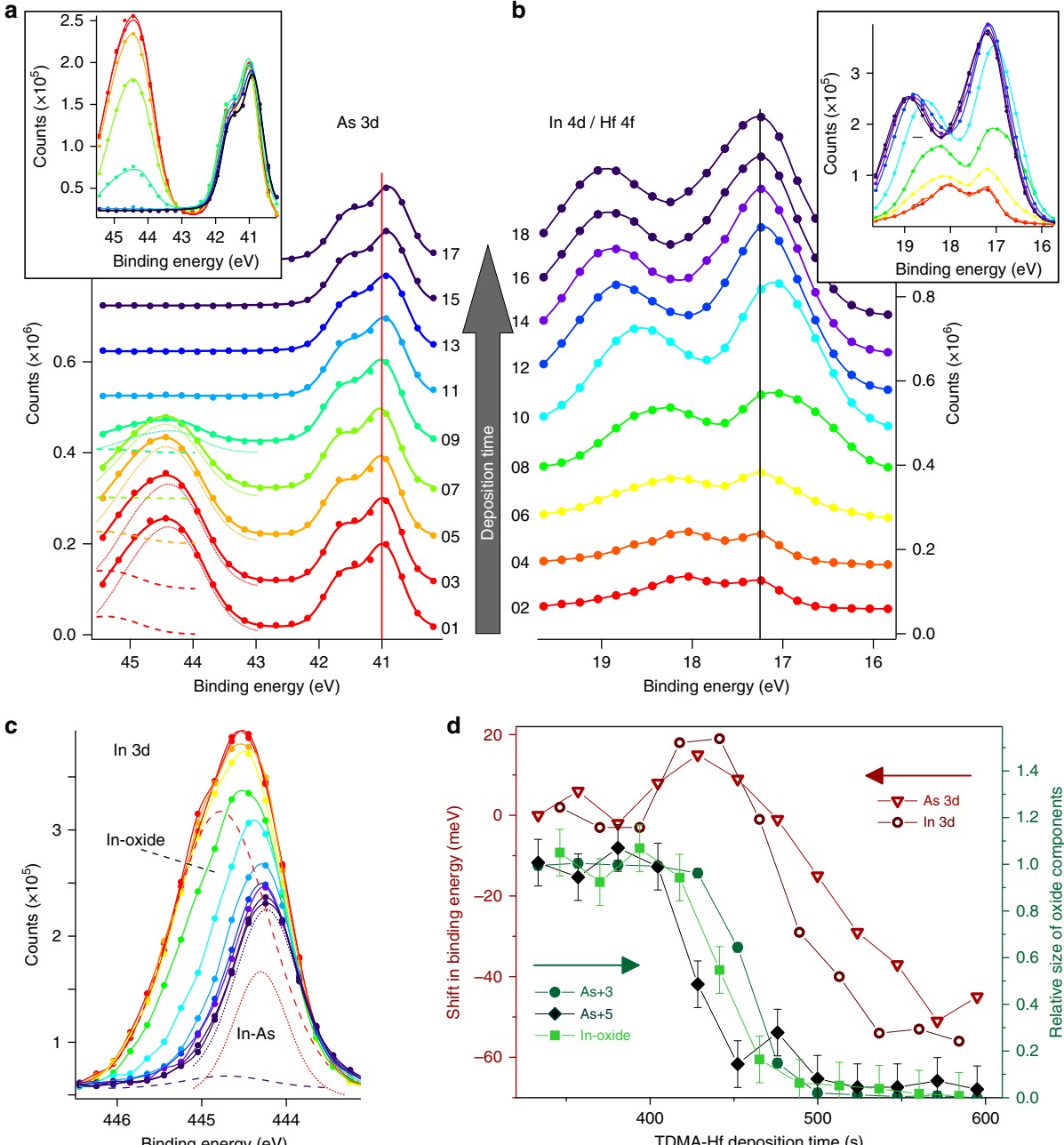

**Fig. 3** Correlation of Hf surface chemistry and oxide removal. Correlated time evolution of (**a**) As 3d and (**b**) In 4d/Hf 4f core level spectra, obtained alternately during TDMA-Hf deposition of the first ALD half-cycle, as indicated by the sweep index labels on each spectrum. Thereby As-oxide reduction and Hf surface chemistry can be correlated for spectra shortly before (red), during (orange, yellow, green, cyan, blue, purple), and after (dark purple) the reaction. The insets show the same spectra without being shifted along the y-axis, for easier comparison of peak heights. Experimental data are plotted by filled circles, while the lines in **a** show fitted spectra. Fitted $As^{3+}$ and $As^{5+}$ oxide components are indicated in **a** by dotted and dashed lines, respectively. Contributions from different overlapping In 4d and Hf 4 f components to the spectra shown in **b** are plotted in Supplementary Fig. 5 and discussed in detail in Supplementary Note 5. Binding energies of the As bulk component before the reaction (red line, **a**) and the Hf–O component after the reaction (dark purple line, **b**) are indicated. A narrow binding energy range has been chosen for the spectra in **a**, **b** in order to minimize the acquisition time and maximize the temporal resolution. **c** In 3d spectra, obtained alternately with As 3d spectra (shown in Supplementary Fig. 6). Fitted In-oxide and In-arsenide (bulk) components are plotted as dashed and dotted curves, respectively, for the first (red) and last (dark purple) spectra. Shifts in the binding energy of the fitted As 3d and In 3d bulk components during the reaction are shown in **d**, together with the change in peak area of the $As^{3+}$, $As^{5+}$, and In-oxide components (with relative peak areas normalized to "1" before the onset of the reaction). Error bars correspond to uncertainties in fitting different oxide components. The time delay between subsequent spectra from the same core level was 20 s in **a**, **b** and 24 s in **c**, **d**. Spectra were obtained at photon energies of 330 eV (**a**, **b**) and 570 eV (**c**, **d**), a gas pressure of about 0.8 Pa, and a sample temperature of 220 °C

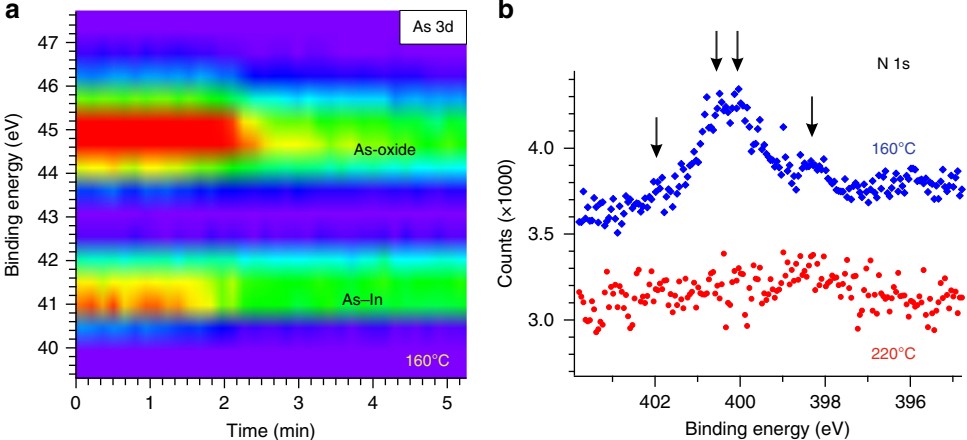

**Fig. 4** Incomplete ALD reaction at low temperatures. **a** Time evolution of As 3d XP core level spectra during exposure to TDMA-Hf at a sample temperature of 160 °C and a gas pressure of about 2 Pa. Components corresponding to As oxide and to As bound to In are indicated. An only incomplete reduction of the oxide component can be observed. **b** Nitrogen (N) 1s core level spectra obtained under UHV conditions after water deposition at sample temperatures of 160 °C (blue curve) and 220 °C (red curve). Separate N components are indicated by arrows. Spectra have been obtained at photon energies of 320 eV (**a**) and 530 eV (**b**)

thickness. Similar results were observed for sample temperatures of both 160 °C (Fig. 4a) and 180 °C (Supplementary Fig. 7). At these low temperatures, different nitrogen species also become incorporated in the surface layer, which is not the case at a temperature above 200 °C (Fig. 4b). Thus, the lower temperature limit for the chemical processes relevant for the removal of the native oxide and the formation of a pure HfO₂ layer is between 180 and 200 °C.

## Discussion

The established view of the chemical reactions during ALD deposition of HfO₂ or other high-*k* oxides on a semiconductor surface is based on a ligand exchange mechanism. First, one or more ligands are split off from the central Hf atom during the first half-cycle. The Hf atom thereby forms a chemical bond with an O atom on the surface, while still being terminated by ligand(s) away from the surface. Then, during the second half-cycle, the remaining ligands are replaced by hydroxyl groups and the process can start over[10,11,13,32]. Our findings show that this picture is too simple to describe the chemical reaction at the surface during the first ALD half-cycle. These insights are a result of our unique possibility to measure not only after each half-cycle (having pumped out all reaction gases), but also during these half-cycles.

The main difference in the established view of the ALD process is our observation of a hitherto unknown two-step reaction involving TDMA-Hf deposition, which manifests as a change in the Hf binding energy during the initial deposition and incorporation of Hf (Fig. 2a, d and Fig. 3b). The most straightforward two-step reaction that could account for this observation is the molecular adsorption of Hf precursor molecules on the InAs surface prior to the main chemical reaction of ligand dissociation and Hf–O bond formation. (We use the term "molecular adsorption", equivalent to "molecular chemisorption", to describe the adsorption of an entire molecule via a donative bond. This is in contrast to "dissociative chemisorption", where a part of the initial molecule is split off and bond reformation occurs. This terminology is common in surface chemistry, while it might be unfamiliar to the materials science community.) Because the oxidation number of the Hf atoms is expected to be +4 both in the TDMA-Hf precursor molecules [N(CH₃)₂]₄Hf and in the HfO₂ film, a simple change of oxidation state cannot explain the increase in Hf binding energy between the first and second step.

Molecular adsorption of TDMA-Hf on Si has been suggested, based on DFT calculations, to occur via a nucleophilic attack of an N-lone pair of electrons in a dimethylamido ligand, which decreases the electron density at the N atom but increases it at the central Hf atom[40]. (An "N-lone pair", following chemistry terminology, corresponds to a "filled dangling bond of an N atom" in physics terminology.) We assume this mechanism to occur also on InAs, where the increased electron density leads to a decreased binding energy of the Hf atoms in the adsorbed TDMA-Hf molecules. The large peak size of the Hf 4f spectra already during this molecular precursor adsorption (green curve in Fig. 2d/ curves 08 and 10 in Fig. 3b, and Supplementary Fig. 5h, i) emphasizes the significance of this first reaction step. As a second step, we expect the dissociation of a dimethylamido ligand from the central Hf atom together with Hf–O bond formation with a surface O atom. During this second step, the XPS binding energy of the Hf atoms is increased toward the typical value corresponding to Hf–O bonds in HfO₂. A number of theoretical models can be found that describe the chemical reactions involved in this second step, such as ligand dissociation and Hf–O bond formation[24,36,41] This step implies an activation barrier that needs to be overcome, as the bond dissociation energy for breaking the Hf–N bond amounts to about 110 kcal/mol, while the newly formed Hf–O bond has a dissociation energy of about 140 kcal/mol, which results in an exothermic reaction[24]. However, our work shows that the picture of the ALD process remains incomplete as long as it does not also include the first step of precursor molecular adsorption, which has often been neglected until now.

The precursor molecular adsorption step is even more important, since our results show that the self-cleaning process (removal of the native oxide) occurs mainly during this part of the reaction. Until now, it was only known that the self-cleaning happens during the first ALD half-cycle[7,21] and the ligand exchange mechanism was supposed to be the driving force, which would imply that the oxygen atoms of the native oxide are converted into HfO₂ oxygen atoms. Theoretical models can well explain the formation of Hf–O bonds upon this ligand exchange mechanism[24,40] and further decomposition of the ligand[41], and some also include the molecular adsorption of precursors onto OH-terminated surfaces upon further ALD cycles[36,40]. However, an explanation of how this reaction upon the initial precursor deposition can remove a 1–2 nm thick layer of In and As oxides[15]

and at the same time produce less than 1 atomic monolayer of $HfO_2$ is still lacking. Some authors reported diffusion of InAs surface oxides through an already deposited $TiO_2$[42,43], or $Ta_2O_5$[44] layer, followed by oxide removal under continuous ALD precursor supply, enhanced by further ligand decomposition[41,42]. However, this effect cannot explain the complete removal of the native oxide already during the first ALD half-cycle reported elsewhere[7,21]. Our time-resolved XPS results, which correlate native oxide removal and Hf surface chemistry (Fig. 3), indicate that the majority of the native oxide has already been removed when the first Hf–O bonds are formed, giving evidence that the oxygen atoms of the native oxide are not directly incorporated into $HfO_2$. Instead, we suggest that the molecular adsorption of the TDMA-Hf precursor is responsible for the self-cleaning process—possibly by forming a bond between the nitrogen atom of a dimethylamido ligand and an oxygen atom of the native oxide, which later leads to desorption of the oxygen atom from the surface together with the organic ligand. Future theoretical studies are needed to further investigate the involved processes.

Another unexpected result of our study is the temporal evolution of the self-cleaning process. The $As^{5+}$ oxide component, which can correspond to $As_2O_5$ or $InAsO_4$, is found to be removed from the surface before the $As^{3+}$ oxide component of $As_2O_3$ (Fig. 3d). However, $As_2O_3$ is energetically less stable than $As_2O_5$ or $InAsO_4$, according to the corresponding Gibbs free energies[45]. This indicates that the order of the native oxide removal is not primarily controlled by thermodynamics. Taking into account that native oxide removal occurs during the precursor molecular adsorption, the different coordination of the oxygen atoms in the native oxide should be considered more important, as the oxygen atoms in $As_2O_5$ seem to be preferential to those in $As_2O_3$ for forming an O–N bond with the precursor ligand.

The non-instantaneous, two-step reaction of the first ALD half-cycle further implies that a sufficiently long ALD pulse length, i.e., duration of precursor deposition, is critical for ALD film growth and interface formation. In principle, the self-limiting chemical reactions of the ALD processes should result in the complete coverage of the surface with Hf-containing molecules upon the first ALD half-cycle, and ideally in the deposition of one complete, homogeneous layer of $HfO_2$ after one full ALD cycle[10,11]. In practice, however, typical ALD $HfO_2$ growth rates are in the range 0.08–0.15 nm/cycle[15,19,46–48] corresponding to less than 0.5 monolayer/cycle. This discrepancy is due to incomplete precursor coverage of the surface because of steric effects[36] to incomplete surface reactions, and to kinetic limitations during the short deposition pulses. In our study we found the average thickness of the $HfO_2$ layer after the first and second full ALD cycle to be about 0.12 nm and 0.37 nm, respectively (see Supplementary Note 6 and Supplementary Fig. 8). These values are within the span of observed ALD growth rates, but near the upper limits. We attribute this to the well-monitored and very long half-cycle pulse

lengths, where the exposure of the surface to the precursor was continued until we could observe no further changes in the surface chemistry by AP-XPS. As a result, we are certain that the full reaction has occurred—which is not necessarily the case in a standard unmonitored situation. We also note that our reactions are still kinetically limited (Supplementary Note 1). Regarding the efficiency of the self-cleaning effect, our study—with long ALD pulses and a relatively high ALD growth rate—shows the complete removal of As oxides and very efficient removal of In oxides. Even a perfect self-cleaning effect will not remove all group-III oxide, since the interface between an InAs (or GaAs) substrate and the high-$k$ layer is formed via In(Ga)–O–Hf bonds[15,49]. Thus, the self-cleaning observed in this study comes close to the ideal case, while many reports on a significant amount of residual interfacial oxides can be found in the literature, which are often connected to rather low ALD growth rates[15,19,46] This goes against the conventional idea that pulse lengths in ALD are rather unimportant due to a quasi-instantaneous and self-limited description of ALD. Here, we point instead to a procedure where it can be beneficial to prolong especially the first ALD half-cycle, both for improving the interface quality and for enhancing ALD film growth.

Our findings also indicate that complete dissociation and desorption of the precursor ligands occurs above a thermal activation barrier, which is between 180 and 200 °C. This results from analyses of our surfaces after water deposition, which show almost no organic residuals (Fig. 4b) if the temperature during deposition is 200 °C or higher (while significant amounts of nitrogen and carbon species are found after deposition at 180 °C or lower). These results are in contrast to the previous works that reported organic species being incorporated in the $HfO_2$ films as by-products of the ligand exchange mechanism, even at higher temperature[25,50]. However, the existence of a thermal activation barrier for the complete desorption of organic residuals from the sample surface agrees with density functional theory calculations: while the initial dissociation of $N(CH_3)_2$ ligands from the TDMA-Hf molecule is found to be strongly exothermic, further dissociation and desorption of these ligands are predicted to occur if sufficient thermal activation energy is provided[41]. At deposition temperatures below the thermal activation barrier, we also found an incomplete removal of the native oxide upon TDMA-Hf deposition (Fig. 4a). For the ALD formation of $HfO_2$ films from TDMA-Hf and water precursors on GaAs, the critical temperature for almost complete native oxide removal was reported to be 250 °C[20] or between 250 and 300 °C[51], which is 50–100 °C higher than for InAs substrates used here.

Although the current experiment has been restricted to the model system of TDMA-Hf and water deposition on InAs, there is no reason to expect a principally different reaction mechanism for other III–V substrates, including the technologically relevant surfaces of (In)GaAs, InSb, or GaSb. Some parameters, such as the thermal activation barrier (as discussed above for GaAs) or

**Table 1 XPS fit parameters**

|  | Spin–orbit splitting (eV) | Branching ratio | Lorentzian FWHM (eV) | Gaussian FWHM (eV) | Asymmetry factor |
|---|---|---|---|---|---|
| As 3d | 0.669 | 1.5 | 0.16 | 0.5–0.7 | 0.00 |
| In 3d | 7.56 | 1.63 | 0.31 | 0.6–0.7 | 0.034 |
| In 4d | 0.861 | 1.56 | 0.221 | 0.5–0.7 | 0.024 |
| Hf 4f | 1.66 | 1.35 | 0.1 | 0.8–1.1 | 0.03 |
| O 1s | / | / | 0.38 | 1.2 | 0.029 |

Parameters used to fit the core level spectra are shown. Spin–orbit splitting, the branching ratio defining the height difference between both components of the doublet, Lorentzian FWHM and asymmetry factor were used consistently for all spectra of the same core level and also for all components within one core level spectrum. The Gaussian FWHM was kept free during fitting, and the values shown here are the results for the bulk components (like the InAs components in the As 3d and In 3d spectra), while oxide components (like $As^{+3}$ oxide, $As^{+5}$ oxide, or In oxide) typically have larger Gaussian FWHMs

the specific timescales of the ALD reaction steps, will definitely vary for different substrates or metalorganic precursors. But we expect that the observed two-step reaction mechanism is valid for all ALD processes on semiconductors that include an energetically favorable adsorption of metalorganic precursor molecules on the native oxide surface and an activation barrier for the ligand dissociation. Further AP-XPS studies are needed to extend this insight in the temporal evolution of ALD reactions on other areas of ALD processes.

This study provides substantial evidence that the efficiency of the self-cleaning process upon ALD, and with that the quality of the resulting interface between the III–V semiconductor and the high-$k$ oxide film, can be controlled by the surface chemistry involved in the molecular adsorption of suitable precursors, rather than the subsequent high-$k$ oxide formation. An almost complete removal of the native oxide has already been reached here by carefully monitoring the ALD process, giving hope for establishing fully defect-free semiconductor/high-$k$ oxide interfaces. Furthermore, this previously unknown mechanism within the ALD chemistry opens up alternative strategies for tailoring ALD precursors according to their molecular adsorption behavior and using special precursor chemistry during the first ALD half-cycle, thereby creating functionalized interfaces providing superior device performance.

## Methods

**InAs sample preparation**. InAs (001) substrates from Wafer Technology Ltd., n-type doped by S atoms, have been cleaned and prepared for the ALD process by a 30 s etch in HCl:H$_2$O 1:1, followed by a 60 s rinse in de-ionized water. After this, samples were blown dry with nitrogen, quickly bonded to sample plates by In-bonding at about 140 °C, and loaded into the UHV chamber of the AP-XPS setup within 5–10 min. Although the etching should in principle result in a clean InAs surface, our previous results[15] have shown that the short exposure of the etched surface to air is sufficient to re-oxidize the sample and create a native oxide layer of about 1 nm thickness on the otherwise clean InAs surface.

**ALD at the AP-XPS setup**. All experiments were performed at the AP-XPS endstation at the undulator beamline I511/SPECIES of the MAX II synchrotron electron storage ring at the MAX IV Laboratory in Lund, Sweden, which allows XPS measurements both under UHV conditions and at pressures up to about 1 kPa. The sample was located in an ambient pressure reaction cell, which was docked to the front aperture of a differentially pumped SPECS PHOIBOS 150 NAP electron analyzer. The reaction cell was operated in a constant flow mode so that the ALD precursor gases were let in through a gas dosing leak valve, and the reaction products were pumped out of the cell by a turbo molecular pump continuously. Inside the reaction cell the sample was heated to temperatures between 160 and 220 °C. The temperature of the sample was monitored by a K type thermocouple, and the absolute accuracy of the measurement is estimated to be approximately 20 °C, while temperature changes were followed with much higher sensitivity. Details about the setup can be found in refs. [29,31,52,].

A TDMA-Hf container (from Sigma-Aldrich Corporation / SAFC Hitech) was attached to the gas system of the setup and heated to about 70 °C in a water quench to reach a sufficient vapor pressure (above 10 Pa). Water vapor was obtained from a small reservoir of de-ionized liquid water at room temperature. The gas flow of either TDMA-Hf or water into the gas supply line of the ambient pressure cell was regulated by hand using separate dose valves. To obtain the vapor pressure inside the ambient pressure reaction cell, the pressures were measured at the first part of the analyzer differential pumping system, separated from the reaction cell by a 0.3 mm aperture. The pressure inside the reaction cell was then determined from the measured value using calibration curves determined earlier during the instrument commissioning. In addition, the gas composition and pressure changes in the reaction cell were monitored by a quadrupole mass spectrometer, which was connected to the gas outlet line of the reaction cell through a leak valve.

After supplying either TDMA-Hf or water to the InAs sample, the gas supply line and the ambient pressure cell were pumped to UHV conditions before the next, alternating deposition was started. Thus, high-resolution XP spectra of various core levels could be obtained in UHV after every ALD half-cycle at photon energies varying between 70 and 1000 eV. During each deposition, usually only one core level was monitored by subsequent AP-XP spectra (in some cases, as in Fig. 3, AP-XP spectra from two core levels were obtained alternately). Thereby a compromise between high repetition rate, resulting in high time resolution, and sufficient energy resolution and count rate had to be found. AP-XP spectra shown here were obtained with repetition times of 11–17 s between subsequent spectra, distances between neighboring acquisition points of 0.15 or 0.2 eV, pass energies of

20 eV (As 3d core level spectra) or 50 eV (In 4d / Hf 4f and O 1s), and photon energies between 170 and 660 eV. The time-resolved AP-XPS maps (Figs. 1a, 2a-c, 4a) have been smoothed for presentation.

In principle, the photoelectrons detected in the AP-XP spectra can come from the sample surface or from the gas phase in the ambient pressure cell. However, the latter contribution only becomes relevant at gas pressures of 100 Pa or higher. Our experiments were performed at a maximum gas pressure of 3 Pa, thus we can neglect any direct contribution from the gas phase in our spectra.

**XPS spectra fitting**. XP core level spectra were fitted using the IGOR Pro and FITXPS software, assuming a Doniach-Sunjic line shape. Fit parameters as given in Table 1 were used consistently for all spectra, in agreement with earlier measurements[15,33]. Binding energies were calibrated using the values of 41 eV for As 3d 5/2 (bound to In), 444.3 eV for In 3d 5/2, 17.3 eV for In 4d 5/2 (both bound to As), and 530.5 eV for O 1s (bound to Hf, without the presence of –OH bonds). The Gaussian full width at half maximum (FWHM) includes the instrumental broadening of the electron analyzer and the photon energy resolution, which varies with the experimental XPS parameters and contributes with a broadening of between 0.2 and 0.9 eV. Detailed values of the instrumental broadening for different XP core level spectra can be found in Supplementary Table 1.

**Data availability**. The data that support the findings of this study are available from the corresponding author upon reasonable request.

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

## Acknowledgements
This work was performed within the NanoLund Centre for Nanoscience at Lund University, and was further supported by the Swedish Research Council (VR), the Swedish Foundation for Strategic Research (SSF), the Crafoord Foundation, the Knut and Alice Wallenberg Foundation, and the European Research Council as well as the European Commission under the European Union's Seventh Framework Programme, Grant Agreements Nos 251862, 259141, and 608153. The authors thank Benjamin Reinecke and Margit Anderson for experimental support, Robin Swärd and Niclas Johansson for assistance in the data analysis, and Dan Csontos for reviewing the manuscript prior to submission. Jesper N. Andersen is gratefully acknowledged for helpful discussions.

## Author contributions
A.M., J.S., J.K. and R.T. conceived the idea and developed the layout of the experiment. M.H. prepared the samples, A.R.H., S.U., and J.K. operated the AP-XPS system, and R.T., S.Y., J.V.K., S.R.M., A.T., and O.P. performed the gas deposition processes and acquired the XP spectra during several synchrotron experiments. R.T., A.R.H., and S.Y. analyzed the data. R.T. prepared the manuscript in collaboration with A.M. All authors discussed the results and commented on the manuscript.
