## [Peer Review File · Nature Communications]

Reviewer #1 (Remarks to the Author):

This manuscript describes the surface interactions and self-cleaning process of InAs during HfO₂ by using a new ambient-pressure XPS. This is a beautiful study with a novel technique that sheds new light on the initial interactions of high-k ALD precursors and the native oxides of III-V semiconductor surfaces. The findings are novel and illustrate that previous models and conclusions were incomplete or wrong due to not having an analysis technique that could show a more granular time scale to the interactions.

I think this manuscript should be accepted as is and will be of major interest to those in the III-V community and in other fields that are utilizing ALD and the self-cleaning process to engineer interfaces.

Reviewer #2 (Remarks to the Author):

In this manuscript the authors investigate the removal of the InAs surface native oxides during the exposure of the surface to the ALD precursor tetrakis dimethylamino hafnium. This is one of the processes for which the self-cleaning reaction has been confirmed. In this work the starting surfaces have been etched but according to the authors about 0.7-1.1 nm of the native oxide still remains. We will call these residual native oxides "surface oxides" in this review. The authors perform their measurements in a specially designed ambient pressure x-ray photoelectron spectroscopy (AP-XPS) set up that allows monitoring of the surface during the actual precursor exposure rather than after each process half cycle as is traditionally done using conventional UHV XPS. Unlike conventional ALD processes the half cycles used by the authors are extremely long, of the order of several minutes. The authors perform conventional UHV XPS measurements after each half cycle.

The authors observe a substantial change in the signal associated with the arsenic oxides after over 2 minutes of surface exposure. The onset of the removal for the surface indium oxides takes even longer. The authors hypothesize that slow molecular adsorption of the precursor is responsible for these removal reactions. They base these assertions on data shown in Figure 3 of the main manuscript and Figure S3 of the supporting information. The data presented in Figures 3a and c is very convincing that the oxide removal does occur. However, this reviewer questions the interpretation of the data in Figure 3b and S3. The authors choose to examine the In 4d region that overlaps significantly with the Hf 4f region making analysis almost impossible. Yes the authors are able to fit the data very well in S3 but they need to use two Hf components Hf I and Hf II. There is actually a much simpler explanation for the shifting of the Hf 4f peaks initially to lower binding energies and then to higher binding energies after the removal of the surface oxides (lines 193-199 main manuscript and Figure 3b). What the authors observe is not the shift of the Hf oxide peak binding energies but the emergence of the Indium substrate peaks. As the authors show in the first red trace of Figure 3b the substrate peak is covered by the ~1 nm residual layer of native (surface) oxide. As a result the In-As peak is attenuated. As the native (surface) oxide is removed this attenuation becomes smaller and smaller. In the end of this very long first half cycle the authors state that they have deposited ~0.12 nm of HfO₂ and removed about 1 nm of combined arsenic and indium oxide. As a result, the signal for the In from the substrate in the 4d region will increase substantially. The authors never consider this in their analysis. In this reviewer's opinion what really causes the native oxide removal is the precursor decomposition after prolonged exposure. Decomposition is not a self-limiting process and when coupled with the fact that both arsenic and indium native oxides are known to be mobile (Ye et al., ACS Appl. Mater. Interfaces 5(16), 8081-8087, (2013), Henegar et al. ACS applied materials & interfaces 8 (3), 1667-1675 (2016), Henegar et al. J. Vac. Sci. Technol. A 34 (3), 031101 (2016)) then that can explain the removal of ~1 nm of native oxides in 20 sec. Amine precursors are known to decompose at 200C and in a recent publication Klejna and Elliott (ref 41 in the current manuscript) already have explained that the ligand exchange mechanism cannot account for the self-cleaning reaction. Rather they propose precursor decomposition as a required step to complete this reaction and the

current manuscript seems to demonstrate that as well.

At minimum, the authors need to examine a much cleaner spectral region for Hf than the Hf 4f which although the strongest, overlaps substantially with the In 4d region. There are other Hf lines that the authors can examine to substantiate their claim about the molecular chemisorbed precursor. Until the authors provide such convincing evidence though, this manuscript is not suitable for publication.

A few minor issues

Figure 3a. The authors cut-off the high binding energy side of the arsenic oxide peak. Why?

Figure 3b. Same as above for the purple spectra. The high binding energy side of the spectrum if cut-off.

Point-to-point response to the referee's comments

Reviewer 1:

This manuscript describes the surface interactions and self-cleaning process of InAs during HfO₂ by using a new ambient-pressure XPS. This is a beautiful study with a novel technique that sheds new light on the initial interactions of high-k ALD precursors and the native oxides of III-V semiconductor surfaces. The findings are novel and illustrate that previous models and conclusions were incomplete or wrong due to not having an analysis technique that could show a more granular time scale to the interactions.

I think this manuscript should be accepted as is and will be of major interest to those in the III-V community and in other fields that are utilizing ALD and the self-cleaning process to engineer interfaces.

Answer:

We are very appreciative towards the reviewer for acknowledging the quality of our work and for stating that the paper is of high relevance.

Reviewer 2

In this manuscript the authors investigate the removal of the InAs surface native oxides during the exposure of the surface to the ALD precursor tetrakis dimethylamino hafnium. This is one of the processes for which the self-cleaning reaction has been confirmed. In this work the starting surfaces have been etched but according to the authors about 0.7-1.1 nm of the native oxide stills remains. We will call these residual native oxides "surface oxides" in this review.

Answer:

We are grateful to the reviewer for noticing us of a possibly confusing detail in the methods section of our manuscript. The etching of the sample should indeed result in a clean InAs surface. However, our previous results have shown that the short exposure of the etched sample towards ambient conditions is enough to re-oxidize the sample, resulting in an about 1 nm thick native oxide (or surface oxide). Nevertheless, the surface is expected to be significantly cleaner than that of the originally purchased sample. We have added a short explanation on this in the revised manuscript (lines 417 – 420).

The authors perform their measurements in a specially designed ambient pressure x-ray photoelectron spectroscopy (AP-XPS) set up that allows monitoring of the surface during the actual precursor exposure rather than after each process half cycle as is traditionally done using conventional UHV XPS.

Answer:

We are again very appreciative to see that also reviewer 2 points out the uniqueness of our approach.

Unlike conventional ALD processes the half cycles used by the authors are extremely long, of the order of several minutes. The authors perform conventional UHV XPS measurements after each half cycle. The authors observe a substantial change in the signal associated with the arsenic oxides after over 2 minutes of surface exposure. The onset of the removal for the surface indium oxides takes even longer. The authors hypothesize that slow molecular adsorption of the precursor is responsible for these removal reactions. They base these assertions on data shown in Figure 3 of the main manuscript and Figure S3 of the supporting information. The data presented in Figures 3a and c is very convincing that the oxide removal does occur.

Answer:

We agree that the oxide removal is the central result of the ALD process.

However, this reviewer questions the interpretation of the data in Figure 3b and S3. The authors choose to examine the In 4d region that overlaps significantly with the Hf 4f region making analysis almost impossible. Yes the authors are able to fit the data very well in S3 but they need to use two Hf components Hf I and Hf II. There is actually a much simpler explanation for the shifting of the Hf 4f peaks initially to lower binding energies and then to higher binding energies after the removal of the surface oxides (lines 193-199 main manuscript and Figure 3b). What the authors observe is not the shift of the Hf oxide peak binding energies but the emergence of the Indium substrate peaks. As the authors show in the first red trace of Figure 3b the substrate peak is covered by the ~1 nm residual layer of native (surface) oxide. As a result the In-As peak is attenuated. As the native (surface) oxide is removed this attenuation becomes smaller and smaller. In the end of this very long first half cycle the authors state that they have deposited ~0.12 nm of HfO₂ and removed about 1 nm of combined arsenic and indium oxide. As a result, the signal for the In from the substrate in the 4d region will increase substantially. The authors never consider this in their analysis.

Answer:

We agree that the reviewer's explanation for the shifting of the Hf 4f peaks sounds intuitive at first view. We therefore performed a **thorough re-analysis of our experimental data, assuming** that the observed shift is not caused by an actual change of the Hf contributions, but by **a change of the In signal** due to

the removal of the native (surface) oxide, as suggested by the reviewer. This detailed analysis can now be found as a **new section 5 of the Supplementary Information**.

The reviewer correctly states the importance of the Indium substrate peaks. We distinguish here between a doublet given by In-As (Indium substrate) and a doublet combining several In-oxides. We find an energy separation between both doublets of 0.6 eV, with the substrate component at lower binding energy. These values are in agreement with our previous results and with literature (e.g. Brennan et al., J. Appl. Phys. 108, 053516, or Martinez et al., Chem. Phys. Lett. 539, 139). We have to point out here that our spectra are obtained at a photon energy of 330 eV, resulting in a kinetic energy of the Hf 4f / In 4d photoelectrons of about 310 eV. At this energy, we expect an inelastic mean free path of the electrons in the native (surface) oxide on InAs of about 9 Å (based on the TPP-equation). Thus, **we should expect to see contributions both from the surface oxide and from the substrate in the In 4d spectra**. Indeed, our fits, shown in Figure S5a and c, result in In-oxide and In-As components of nearly the same height. It is important to consider that the surface oxide consists of a mixture of As- and In-oxides (with roughly 65% As-oxides and 35% In-oxides). **Removal of these oxides will therefore not lead to a significant increase of the total In signal** – instead, we expect the In-As signal to increase in roughly the same extent as the In-oxide signal decreases. This will lead to some shift of the binding energy in the resulting In 4d spectra towards lower BE upon oxide removal, but the total intensity of the In 4d spectra is not expected to increase by more than at maximum a factor two. In fact, the AP-XP **In 3d spectra**, which are monitored during the first deposition of TDMA-Hf, even show a decrease in total intensity as the reaction sets in (**Figure 3c**). This is a **clear indication that the total In contribution to the XPS intensity does not increase during the self-cleaning reaction**. This behavior is also reflected by the energy separation of the doublet peaks in the overlapping Hf 4f / In 4d spectra obtained during the reaction: When the total intensity of the spectra increases, also the shape changes from that of an In 4d doublet (spin-orbit splitting of 0.86 eV) towards that of an Hf 4f doublet (spin-orbit splitting of 1.65 eV), indicating that the additional photoelectrons arise from Hf atoms. This can best be followed in the inset of Fig. 3b.

Nevertheless, **we tried to model the experimental data of Fig. 3b by an increase of the In substrate signal, and alternatively also by the emergence of additional In components, following the suggestion of the reviewer**. The results of this alternative data analysis are shown and discussed in detail in the new section 5 of the SI, including Figure S5. However, following this assumption that the Hf peak does not shift but that the observed shift is due to changes in the In signal, we **could not successfully fit the spectra obtained during the reaction (Figure S5e,f)** due to **three reasons**: (1) The **shift in binding energy** between In-oxide and In-As (substrate) components is not large enough to explain the observed shift. **We would need to introduce an extra In component at 0.6 eV below the binding energy of the In substrate peak** in order to reach the low binding energy tail that is present in our data. Such a strong negative chemical shift has to our knowledge never been reported in the InAs material system. (2) In order to explain the observed shift in binding energy during the chemical reaction by the emergence of In components, **we would need to fit an In contribution which is four times larger than the sum of the In-oxide and In-As (substrate) peaks before the onset of the reaction**. Such a large increase cannot be explained by the attenuation due to As-oxides, as discussed above. (3) In 4d and Hf 4f doublets have

significantly different spin-orbit splitting (0.86 eV compared to 1.65 eV). **The experimentally obtained spectra show a very clear signature of a Hf 4f doublet**, and cannot be fitted by In 4d doublets with the significantly smaller spin-orbit splitting. (This is most prominent for the bright blue sweep #10 of Fig. 3b, which is fitted in Fig. S5f.)

We therefore **conclude that the alternative explanation suggested by the reviewer, although intuitive, cannot be valid here. Instead, our own explanation as discussed in the original manuscript leads to an excellent fit of the experimental data** (Fig. S5h-j).

We want to point out that in addition to the fits shown in Figure S5e-g, we tried various other configurations (using two, three, or more In doublets, with both constrained or free binding energies, varying the branching ratio or the asymmetry factor, testing various starting configurations of the fitting procedure, ...), without any success. The large intensity at lower binding energies, and the large energy separation between the two main peaks in the experimental data, make it impossible to fit the data within the assumption suggested by the reviewer. The applied fitting software, fitXPS, is especially useful for datasets with relatively few measurement points as well as for distinguishing overlapping peaks, since it allows bounds on parameter values and linear constraints between parameters, as we now explain in more detail in section 5 of the SI. Fitting results were also double-checked using IGOR. It should also be mentioned that the relatively small number of measurement points in the AP-XP spectra does not significantly influence the outcome of the fitting procedure, as a comparison of the AP-XP spectrum in Figure S5c and the high-resolution UHV XP spectrum in Figure S5a shows.

We included a short discussion of the two alternative explanations for the observed shift in binding energy in the revised manuscript (lines 154 – 158), together with a reference to section 5 of the SI. Also in the caption of Figure 3, we refer now to the detailed data analysis presented in the SI (lines 215 – 217).

In this reviewer's opinion what really causes the native oxide removal is the precursor decomposition after prolonged exposure. Decomposition is not a self-limiting process and when coupled with the fact that both arsenic and indium native oxides are known to be mobile (Ye et al., ACS Appl. Mater. Interfaces 5(16), 8081–8087, (2013), Henegar et al. ACS applied materials & interfaces 8 (3), 1667–1675 (2016), Henegar et al. J. Vac. Sci. Technol. A 34 (3), 031101 (2016)) then that can explain the removal of ~1 nm of native oxides in 20 sec. Amine precursors are known to decompose at 200C and in a recent publication Klejna and Elliott (ref 41 in the current manuscript) already have explained that the ligand exchange mechanism cannot account for the self-cleaning reaction. Rather they propose precursor decomposition as a required step to complete this reaction and the current manuscript seems to demonstrate that as well.

Answer:

We agree that the studies by Ye et al. and Henegar et al. provide important additional insight into the complex behavior of oxide removal during the ALD process, and we therefore refer to these studies in

the revised manuscript (lines 323 – 326). In those studies, the authors observe the post-deposition removal of interface oxide on InAs by the diffusion of In-oxides and As-oxides through a several nm thick TiO₂ or Ta₂O₅ layer deposited by ALD. However, the situation is different in the current study, where most of the In-oxides and all As-oxides are removed from the surface before the first ALD half-cycle is completed (which results in a film thickness of only about 1 Å). In a different study from the same group (L. Ye and T. Gougousi, Appl. Phys. Lett. 105, 121604 (2014), ref. [20] of this manuscript), the authors use TDMA-Hf for depositing HfO₂ on GaAs after growing a chemical oxide layer by immersion in H₂O₂. In this case, the strongest removal of As-oxide (from the comparably thick chemical oxide layer) is reported to occur already during the first ALD cycle. This gives strong evidence that the TiO₂ or Ta₂O₅ ALD material systems mentioned above might have different oxide removal mechanisms than the ALD of HfO₂ from TDMA-Hf and water, which is the same process as used here.

While we agree that precursor decomposition is an important factor for the process of native oxide removal, we cannot follow how the reviewer interprets the work by Klejna and Elliott (ref [41]): Klejna and Elliott write that the “*main emphasis here is on the interaction of the dimethylamido ligand (dma) with the oxide surface [...]. We postulate that formation of the dielectric oxide is the driving force for the clean-up effect.*” This is fully in line with the ligand exchange model. The authors describe the “*clean-up performance*” by the equation

(1): “ $\text{||As-O||} + 1/n \text{ML}_{n(g)} \rightarrow \text{||O-M}_{1/n} + \text{||As} - \textit{intermediates}$ ”, where M is the metal (Hf).

Later on, Klejna and Elliott continue: “*3.3 Decomposition of dma Ligand. For the following consideration of decomposition processes of alkylamide ligand on the example of dimethylamide we use the highly stable product A ||As-N(CH₃)₂, as a starting species for further clean-up transformations B-K.*”

Accordingly, Klejna and Elliott assume that any decomposition process, leading to intermediate states and transformations relevant for clean-up, occurs *after* the initial ligand exchange process, and therewith *after* the formation of Hf-O bonds.

This is in contrast to this study, where we observe the most relevant part of the oxide removal to occur *prior* to the formation of Hf-O bonds.

At minimum, the authors need to examine a much cleaner spectral region for Hf than the Hf 4f which although the strongest, overlaps substantially with the In 4d region. There are other Hf lines that the authors can examine to substantiate their claim about the molecular chemisorbed precursor. Until the authors provide such convincing evidence though, this manuscript is not suitable for publication.

Answer:

The aim of this study is to monitor the self-cleaning reaction while it is happening, which requires sufficiently fast data acquisition (even if the reaction has been slowed down in our case). In order to obtain as short XPS acquisition times as possible, we need to obtain spectra with high intensities and investigate core levels with as large photoionization cross-section as possible. That’s why we chose to monitor As 3d, In 4d, and Hf 4f core levels. In the accessible photon energy range (above 100 eV), the

cross section of any other Hf core level is at least a factor 10 smaller than that of Hf 4f, as shown in the figure below:

Figure: Photoionization cross-section of Hf core levels, after J. J. Yeh and I. Lindau, *Atomic Data and Nuclear Data Tables* 32, 1 (1985) Reprinted with permission from Elsevier.

Furthermore, there are **tools to clearly distinguish between Hf 4f and In 4d contributions in XP spectra** as discussed above, mainly their significantly different spin-orbit splitting energy and physical constraints such as strong changes in intensity, which have to be used together with a robust fitting procedure. We are confident that **our additional data analysis, presented in Figure S5 and section 5 of the SI, succeeds in non-ambiguous deconvolution of the spectra**. This additional data analysis provides **convincing evidence for the occurrence of a change in Hf binding energy** (due to the precursor molecular adsorption step), and that the alternative interpretation suggested by the reviewer cannot explain our experimental results.

We will continue to investigate this exciting topic and material system. With the given increase in intensity at the SPECIES beamline after relocating to the new MAX IV synchrotron, we will hopefully be able to investigate additional spectral regions, including Hf core levels with lower cross sections. (The SPECIES beamline is the only AP-XPS beamline worldwide that operates a flow reaction cell.) However, as both reviewers have pointed out the relevance of our results, we do not want to wait for additional

results from future synchrotron experiments, but publish our work as soon as possible by this re-submission.

A few minor issues

Figure 3a. The authors cut-off the high binding energy side of the arsenic oxide peak. Why?

Figure 3b. Same as above for the purple spectra. The high binding energy side of the spectrum is cut-off.

Answer:

For the *in-situ* AP-XP spectra, always a compromise between energy resolution, energy range, and acquisition time has to be found. Here we tried to keep the acquisition time as short as possible, ensuring good temporal resolution, therefore we chose a narrow range in binding energy. We agree that the high energy tail of some spectra is cut off, but the focus was set on the shift in binding energy of the main peak, and on the removal of the As-oxide component – both effects can clearly be seen. We have added a short explanation in the figure caption (lines 218 – 220).

By comparing the larger range, high-resolution UHV spectrum of Figure S5a with the narrow range, AP-XP spectrum of Figure S5c (both showing the In 4d signal before the self-cleaning reaction sets in), one can see that the different energy ranges have no relevant effect on the outcome of the fitting procedure and thus on the interpretation of the results.

Reviewer #2 (Remarks to the Author):

In this revised version of the manuscript the authors have attempted to reprocess some of the data to provide evidence for their assertion of the two Hf states present in the film. As I stated in my original review I really believe the authors data and appreciate the time they have invested in setting up the equipment for the experiment. However, I still find their data analysis flawed and as a result cannot in good conscience recommend publication of this manuscript.

More specifically:

In Figure S5 of the supporting documentation the authors have attempted to reprocess presented in Figure 3b of the main manuscript.

They start by assuming that in Figure S5b and d there is mainly (almost exclusively) Hf contribution. It is not clear how thick the HfO₂ film is in this case (not stated) but the spectrum in Figure S5d is recognized in the legend trace 14 of Figure 3b. According to Figure S8 the thickness of the HfO₂ layer after the 1st TDMAT dosing is less than 2 Å. So there should be a significant contribution of the In substrate peaks. Actually the bulk of the peak should be assigned to In rather than Hf. The IMFP for the energy stated (330 eV) should be of the order of 7-8 Å. As a result of this flawed assumption the authors derive a binding energy for Hf 4f of 17.4 which is high for most of the HfO₂ literature. Granted there is spread in the values and there are references that place the BE for Hf 4f in HfO₂ above 17 eV but most references (NIST xps database for example and XPS Handbook) settle at values below 17 eV. In fact the authors state that they need to add a peak at 16.7 eV to fit the data which is precisely the BE where HfO₂ should be. There is an easy way to settle this issue by using a sufficiently thick HfO₂ film.

You really cannot derive any sound conclusions fitting this #14 peak with three doublets within 0.6 eV BE and especially since two of the three are placed within 0.1 eV. I understand the authors argument about cross sections but the signal in the Hf 4d region is only about 1/3 of the Hf 4f (XPS Handbook Moulder et al) which makes it easily suitable for monitoring. Taking the spectra in this much cleaner region (using the 570 eV for example that is available to the authors) beamline should provide convincing evidence either way.

Regarding the commentary on the computational paper from Klejna and Elliot I would like to quote the following sections:

"The first principles study shows that clean-up with metal alkylamides has a similar mechanism to clean-up with metal methyls as regards the scavenging of oxygen from weak As, Ga, and In oxides. Arising from this, ligand exchange can in principle lead to a clean-up product: tris(dimethylamino)arsine. However steric hindrance and the bulky character of the alkylamide ligand are rate limiting factors, which in this case are very pronounced and suggest that this particular reaction will not proceed".

"In the case of the alkylamide ligand, thermal decomposition rather than migration of the entire ligand on the oxide surface is dominant, taking into account the bulky character of the ligand and its known reactivity in contact with a semiconductor or metallic surface. Clean-up of the reducible As oxide substrate is therefore enhanced by secondary decomposition surface reactions, not by oxidation of the entire alkylamide."

In summary while I appreciate the authors effort I cannot recommend publication of this manuscript until the issues identified above have been resolved.

Detailed response to the second referee report

Reviewer #1:

Reviewer #1 concluded *“I think this manuscript should be accepted as is and will be of major interest to those in the III-V community and in other fields that are utilizing ALD and the self-cleaning process to engineer interfaces”* already in the first report.

Reviewer #2 (Remarks to the Author):

In this revised version of the manuscript the authors have attempted to reprocess some of the data to provide evidence for their assertion of the two Hf states present in the film. As I stated in my original review I really believe the authors data and appreciate the time they have invested in setting up the equipment for the experiment. However, I still find their data analysis flawed and as a result cannot in good conscience recommend publication of this manuscript.

More specifically:

In Figure S5 of the supporting documentation the authors have attempted to reprocess presented in Figure 3b of the main manuscript.

They start by assuming that in Figure S5b and d there is mainly (almost exclusively) Hf contribution.

Here we have to contradict: **Our data analysis is free from any assumption regarding the distribution between In and Hf contributions, neither do we define a binding energy for the Hf peak.** So this main point made by the referee is incorrect, leading to a clear misunderstanding of the main point of our response, and of the discussion in chapter 5 of the SI.

In contrary, the ratio between the Hf peak and the In peak as well as the peak energy of the Hf peak result only from the optimization procedure of the peak fitting. Even if we use a start configuration where the In 4d peak is two times as large as the Hf 4f peak and where the Hf 4f peak is centered at 16.7 eV, we obtain the same results as shown in Fig. S5b and d. To further disprove the claim of the referee, we have even attempted to use the central claim of the referee namely that there must not exist a second Hf doublet. Our data re-analysis (Fig S5 d-g) shows unambiguously that the experimental findings cannot be explained within the assumptions of the referee.

The only constraints that we have used for fitting the data of Fig. S5b were the peak parameters of the In 4d doublet (spin-orbit splitting, Lorentzian width, and asymmetry factor), which were obtained before by fitting the data shown in Fig. S5a. All other parameters were found and optimized through the fitting procedure. When working on this revision, we noticed that it is however more important to be consistent through the entire experiment, then to keep the analysis of this specific dataset entirely unbiased, therefore we now applied those peak parameters (spin-orbit splitting, branching ratio, Lorentzian width, and asymmetry factor) that had been optimized before and stated in the method section of the main manuscript, and re-did the fitting procedure for the data of Fig. S5. As a result, the contribution of the In 4d doublet to the spectra of Fig. S5 b and d is now slightly larger, all other changes are marginal.

It is not clear how thick the HfO₂ film is in this case (not stated) but the spectrum in Figure S5d is recognized in the legend trace 14 of Figure 3b. According to Figure S8 the thickness of the HfO₂ layer after the 1st TDMAT dosing is less than 2Å. So there should be a significant contribution of the In substrate peaks. Actually the bulk of the peak should be assigned to In rather than Hf. The IMFP for the energy stated (330 eV) should be of the order of 7-8 Å.

We agree with the referee in that we expect the HfO₂ film to be less than 2 Å thin in this case, and that the inelastic mean free path (IMFP) of the photoelectrons for the given conditions should be in the order of 7 – 8 Å. Since the XPS signal is dominated by contributions from the surface and decays exponentially for contributions from further inside the sample, such a short IMFP results in the top layer (HfO₂) to contribute very significantly, while the influence of the InAs material underneath gets significantly attenuated already by a very thin top layer.

Importantly, the referee forgets to take into account that the photoionization cross section of Hf 4f for the given photon energy is 7 to 8 times larger than that of In 4d (4,979 Mbarn for Hf4f and 0,671 Mbarn for In 4d, at a photon energy of 300 eV, according to Yeh and Lindau, Atomic Data and Nuclear Data Tables 32, 1 (1985)). Thus the signal from the Hf layer will be amplified by a factor of 7-8 compared to the In signal.

Modelling including the IMFP and using photoionization cross sections is found in standard textbooks on the subject, we refer to D. P. Woodruff and T. A. Delchar: “Modern techniques of surface science”, Cambridge Univ. Press, 1994, or Stephan Hüfner: “Photoelectron Spectroscopy – Principles and Applications”, Springer, 2003.

Thus, we do expect a much stronger contribution from the thin HfO₂ layer than from the attenuated InAs substrate (In 4d). This is also in perfect agreement with the in-situ results, which show a significant increase of the total intensity in the In 4d / Hf 4f spectra as soon as the first Hf atoms can be found on the InAs surface (Fig. 2a of the main manuscript).

As a result of this flawed assumption the authors derive a binding energy for Hf 4f of 17.4 which is high for most of the HfO₂ literature. Granted there is spread in the values and there are references that place the BE for Hf 4f in HfO₂ above 17 eV but most references (NIST xps database for example and XPS Handbook) settle at values below 17 eV.

We don't agree with this statement about the binding energy for Hf 4f in HfO₂. The referee mentions the NIST database and the XPS Handbook. The NIST database cites three values from two different sources: 16.7 eV, 17.93 eV, and 18.13 eV (Sarma and Rao, J. Electron Spectrosc. Relat. Phenom. 20, 25 (1980); Morant et al., Surf. Interface Anal. 16, 304 (1990)), which clearly cannot be summarized to “settle at values below 17 eV”. The XPS Handbook by Moulder et al. only cites the older article by Sarma and Rao with the value of 16.7 eV. All other more modern publications that we are aware of mention values between 17.0 and 18.2 eV, including publications from highly reputable XPS groups such as Prof. Wallace at UT Dallas (17.7 eV, APL 102, 131602 (2013)), Kwo and Hong from the Tsing Hua University and National Synchrotron Taiwan (17.7 eV, APL 94, 052106 (2009)), X. Chen from the Chinese Academy of Sciences (17.3 eV, Surf. Sci. Rep. 68, 68 (2013)), Suzer and Hussain at the Advanced Light Source (18.2 eV, J. Vac. Sci. Technol. A 21, 106 (2002)) and many more: 17.0 eV (Pi et al., APL 104, 042904 (2014); Umezawa et al., Jpn. J. Appl. Phys. 6, 3507 (2007)), 17.3 eV (Duan et al., APL 99, 012902 (2011)), 17.5 eV (Platt et al., Th. Sol. Films 518, 4081 (2010)), 17.8 eV (Kang et al., ACS Appl. Mat. Interf. 5, 1982 (2013)). Thus, the vast majority of literature values confirms a BE of more than 17 eV or higher.

In fact the authors state that they need to add a peak at 16.7 eV to fit the data which is precisely the BE where HfO₂ should be. There is an easy way to settle this issue by using a sufficiently thick HfO₂ film.

A central point that should be taken into account here is that the spin-orbit splitting of Hf 4f is very different from In 4d. Thus every In or Hf component must consist of two peaks separated by a well known energy (basic atomic physics). The fit where we mention a component at 16.7 eV was made with an Indium component with its spin-orbit splitting. We can easily test if the suggestion of the

referee of a Hf peak with energies below 17 eV can be accommodated by our data when using the spin-orbit splitting of Hf 4f (which is undisputed). For this test we fit the data of Fig. S5b with the constraints that the Hf peak energy must be below 17 eV, and that the area under the In peak must be larger than that under the Hf peak (as claimed by the referee). The outcome of this attempt is shown to the left, demonstrating that the fit fails completely (see the discrepancy between the experimental data

(black dots) and the best fit (red curve)): The experimental data cannot be fitted under these assumptions (mainly because the energy distance between the two large peaks in the experimental data agrees with the spin-orbit splitting of Hf 4f, but not with that of In 4d). This again proves that the assumptions postulated by the referee do not agree with the experimental data.

You really cannot derive any sound conclusions fitting this #14 peak with three doublets within 0.6 eV BE and especially since two of the three are placed within 0.1 eV.

There is an overwhelming amount of literature where exactly this is done successfully, see for example Brennan and Hughes, JAP 108, 053516 (2010), Tallarida et al., APL 99, 042906 (2011), Huang et al., APL 101, 212101 (2012), Tuominen et al., APL 106, 011606 (2015), or Mäkelä et al., Appl. Surf. Sci. 329, 371 (2015). Such studies are usually not possible with lab-based X-ray sources, but the enhanced energy resolution of a synchrotron beamline enables clear identification of the peaks. This is even more valid in our case, where the doublets come from two different core-levels with significantly different spin-orbit splitting, which makes them clearly distinguishable upon fitting.

I understand the authors argument about cross sections but the signal in the Hf 4d region is only about 1/3 of the Hf 4f (XPS Handbook Moulder et al) which makes it easily suitable for monitoring.

This is unfortunately only true for a lab X-ray source, while in our case a synchrotron source was used: The referee refers to the XPS Handbook, which explicitly shows XPS spectra obtained with X-rays from an Al K alpha lab source (with a fixed photon energy of 1.5 keV). At that photon energy the cross sections are generally much smaller than the ones used in our study, and at that energy also the difference between the cross-sections of Hf 4d and Hf 4f is small (see the region marked green in the figure below). However, in our experiment we tune the photon energy (only possible at a synchrotron) in order to reach the highest possible cross sections and to measure different core levels always with the same electron kinetic energy. If we compare the cross sections at photon energies that result in an electron kinetic energy of 300 eV (as in this case), the cross section is about seven times larger for Hf 4f, as compared to Hf 4d even in the most favorable case, as can be seen in the figure below (indicated by red circles).

Figure: Calculated photoionization cross-sections for Hf core levels, from Yeh and Lindau, *Atomic Data and Nuclear Data Tables* 32, 1 (1985). Hf 4f and 4d cross sections at 320 eV and 520 eV (resulting in a kinetic energy of 300 eV) as well as at 1500 eV are highlighted. Reprinted with permission from Elsevier.

Taking the spectra in this much cleaner region (using the 570 eV for example that is available to the authors) beamline should provide convincing evidence either way.

We always strive to take spectra with the peaks clearly separated, given that we have been doing XPS measurements on these systems for years prior to the present study. However, due to the much smaller cross section of Hf 4d also at 570 eV, we have found that only marginal additional information can be expected from such measurements given the constraints on measurement times in dynamic systems. Instead, proper data analysis must be the way forward here, which is also appreciated by referee 1 who already recommended publication of the manuscript in its original version.

Regarding the commentary on the computational paper from Klejna and Elliot I would like to quote the following sections:

"The first principles study shows that clean-up with metal alkylamides has a similar mechanism to clean-up with metal methyls as regards the scavenging of oxygen from weak As, Ga, and In oxides. Arising from this, ligand exchange can in principle lead to a clean-up product: tris(dimethylamino)arsine. However steric hindrance and the bulky character of the alkylamide ligand are rate limiting factors, which in this case are very pronounced and suggest that this particular reaction will not proceed".

"In the case of the alkylamide ligand, thermal decomposition rather than migration of the entire ligand on the oxide surface is dominant, taking into account the bulky character of the ligand and its known reactivity in contact with a semiconductor or metallic surface. Clean-up of the reducible As oxide substrate is therefore enhanced by secondary decomposition surface reactions, not by oxidation of the entire alkylamide."

We agree with the work by Klejna and Elliott, but we can only repeat the statement from our previous response that the intention of the work by Klejna and Elliott was to investigate what is happening *after* the initial ligand exchange process – they themselves speak about “secondary decomposition surface reactions” in the passage cited above. The main finding of our work describes effects taking place *before* that ligand exchange process. Thus we consider the computational study by Klejna and Elliott as helpful, but not primarily relevant for our work.

In summary while I appreciate the authors effort I cannot recommend publication of this manuscript until the issues identified above have been resolved.

First, we want to point out that we are grateful to the referee for several very helpful suggestions in his/her first report, which we followed upon revising the manuscript. However, we notice that the main point of our response was not considered by the referee. Furthermore, we conclude that the arguments listed by the referee in his/her second report rely on statements that are only valid under different circumstances. Therefore we appeal against the referee’s recommendation, which we consider as unjustified.

Reviewer #2 (Remarks to the Author):

In light of the authors detailed response to the reviewer's comments this reviewer feels that the manuscript should be made available to the community for its consideration. Therefore I recommend publication.

My only remaining request is that the authors make clear in the figure caption(s) the thickness of the HfO₂ film in question (Figure S5d). It would be helpful to the readers if part of the response "We agree with the referee ...(Fig. 2a of the main manuscript)" is incorporated in the manuscript.

Reviewer #3 (Remarks to the Author):

This manuscript presents an evidence of a Hf-absorbed state on the surface of InAs during ALD HfO₂ deposition before HfO₂ formation, which provides new insight and better understanding on the self-cleaning of HfO₂. In-situ AP-XPS measurements using synchrotron as source were carried out, possible mechanisms were proposed by fitting the XPS data. Time-evolution XPS data and different situation before ligands exchange on the surface of InAs were discussed. Overall, this experiment is organized and comprehensive. There are some good comments presented with the referee #2, however, the analysis in this study has been carried out with synchrotron and it causes some arguments between the author and the referee.

In my personal view, the authors carefully study and response the comments from the referee #2 with reasonable explanations. The finding in this report, regarding to in ALD deposition on iii-v semiconductors, is helpful for following research. Using synchrotron and in-situ XPS equipment also demonstrate promising techniques to explore the unknown mechanism of ALD deposition on various substrates. I would like to recommend acceptance of this works with following reasons:

1. As the authors explained, it is possible that a strong XPS signal could be observed with the reduced thickness because the photoionization cross section and thickness should be both considered for the intensity issue. Also, the authors provide sufficient references to support the claim that photoionization cross section is more dominated compared to the thickness.

The referee part:

Due to the penetration depth of the XPS, the intensity of the signal should be dependent on the thickness of the HfO₂. Therefore, the intense intensity might be not consistent with the reduced thickness of initial grown HfO₂ (less than 2 Å in the first half cycle). Disagrees with Figure S5 (b and d) that the peak intensity is mainly attributed to Hf contribution due to that fact that thickness of the HfO₂ in the first half cycle is less than 2 Å and the inelastic mean free path(IMFP) is 7-8 Å for 330eV so that there should be a significant amount of signal comes from the surface of InAs rather than HfO₂.

The authors part:

Explain the issue of photoionization cross section of elements and responds to the referee with the different photoionization cross section of elements that result in such outcome. And thus the author reasonable attribute the signal to Hf dominated.

2. 17.4eV, the binding energy of HfO₂ reported in this paper, is reasonable, which is supported with database of 3 different references.

The referee part:

The inconsistency of reported binding energy in this paper. The reported binding energy of HfO₂ in this paper is higher than most of the reported HfO₂ binding energy. So, the energy used to fit the

data in figure S5 is incorrect.

The authors part:

The authors provide references to support the claim that the binding energy can indeed range from 17eV to 18eV.

3. Hf 4f will have two doublets due to spin-orbital splitting: one is 4f 5/2 which will have higher energy around 19.64 eV while 4f 7/2 which will have lower energy range from 16.7eV to 18.13eV as the author suggested. The assignment of higher energy peak for Hf 4f 7/2 peak is reasonable. It might be not fair to conclude that the fitting of figure S5 is flawed. It might underscore the discovery of this new Hf state during ALD deposition.

The referee part:

The inconsistency of peak assignment of 17.4eV to HfO₂. Where 16.7 eV should be HfO₂'s Binding energy instead of as the author's claim: the "extra In peak". And the referee believes that it is due to this flawed assignment that leads to the need for a new surface state of the Hf.

The authors part:

From the references (NIST Database), the Hf 4f peak indeed can exceed the 17eV limit. And the author also tries to fit the data with the constrained condition that provided by the referee (Using 16.7eV as the HfO₂ peak position), however, the outcomes that uses this boundary condition cannot match with the experimental data.

4. This paper presents new insight for the ALD process by using in-situ measurement with synchrotron source

The referee part:

Data fitting erroneous of the author, where peak positions of the fittings are placed too close to each other. And suggesting the author to monitor the signal of Hf 4d instead of Hf 4f, for that Hf 4d is more suitable for monitoring.

The authors part:

The condition in which the referee believes only held true for the lab X-ray source

5. The reported work (by Klejna and Elliot) mainly focus on the interaction between ligand exchange with oxide, and I quote from the same Klejna and Elliot paper:

" The main emphasis here is on the interaction of the dimethylamido ligand (DMA) with the oxide surface with a view to finding potential redox properties of this ligand. "

In the paper, the authors focus on the adsorption of Hf-containing molecule on the surface before ligand exchange on the surface.

The referee part:

The mechanism proposed by the authors is flawed. With the computational paper from Klejna and Elliot, the referee believes that the dominated factor would be decomposition of ligand on the surface rather than the migration of the whole molecule.

The authors part:

It might not have the same circumstance as described in this report.

Detailed response to the reviewers' comments

Reviewer #2 (Remarks to the Author):

In light of the authors detailed response to the reviewer's comments this reviewer feels that the manuscript should be made available to the community for its consideration. Therefore I recommend publication.

We are delighted to read that our detailed response could dispel the reviewers' doubts about our data analysis.

My only remaining request is that the authors make clear in the figure caption(s) the thickness of the HfO₂ film in question (Figure S5d). It would be helpful to the readers if part of the response "We agree with the referee ...(Fig. 2a of the main manuscript)" is incorporated in the manuscript.

We have added the following sentence in the figure caption of Supplementary Figure 5: "The thickness of the Hf-containing layer deposited by ALD is expected to be less than 0.2 nm." In addition, we have incorporated a substantial part of the mentioned response, i.e. a discussion of the expected thickness of the Hf-containing layer, the photoelectron inelastic mean free path, the photoionization cross-section, and the resulting ratio in XPS signals from In and Hf core-levels, in the first paragraph of Supplementary Note 5.

Reviewer #3 (Remarks to the Author):

This manuscript presents an evidence of a Hf-absorbed state on the surface of InAs during ALD HfO₂ deposition before HfO₂ formation, which provides new insight and better understanding on the self-cleaning of HfO₂. In-situ AP-XPS measurements using synchrotron as source were carried out, possible mechanisms were purposed by fitting the XPS data. Time-evolution XPS data and different situation before ligands exchange on the surface of InAs were discussed.

Overall, this experiment is organized and comprehensive. There are some good comments presented with the referee #2, however, the analysis in this study has been carried out with synchrotron and it causes some arguments between the author and the referee.

In my personal view, the authors carefully study and response the comments from the referee #2 with reasonable explanations. The finding in this report, regarding to in ALD deposition on iii-v semiconductors, is helpful for following research. Using synchrotron and in-situ XPS equipment also demonstrate promising techniques to explore the unknown mechanism of ALD deposition on various substrates. I would like to recommend acceptance of this works with following reasons:

- 1. As the authors explained, it is possible that a strong XPS signal could be observed with the reduced thickness because the photoionization cross section and thickness should be both considered for the intensity issue. Also, the authors provide sufficient references to support the claim that photoionization cross section is more dominated compared to the thickness.*

The referee part:

Due to the penetration depth of the XPS, the intensity of the signal should be dependent on the thickness of the HfO₂. Therefore, the tense intensity might be not consistent with the reduced thickness of initial grown HfO₂ (less than 2Å in the first half cycle). Disagrees with Figure S5 (b and d)

that the peak intensity is mainly attributed to Hf contribution due to that fact that thickness of the HfO₂ in the first half cycle is less than 2Å and the inelastic mean free path(IMFP) is 7-8 Å for 330eV so that there should be a significant amount of signal comes from the surface of InAs rather than HfO₂.

The authors part:

Explain the issue of photoionization cross section of elements and responds to the referee with the different photoionization cross section of elements that result in such outcome. And thus the author reasonable attribute the signal to Hf dominated.

2. 17.4eV, the binding energy of HfO₂ reported in this paper, is reasonable, which is supported with database of 3 different references.

The referee part:

The inconsistency of reported binding energy in this paper. The reported binding energy of HfO₂ in this paper is higher than most of the reported HfO₂ binding energy. So, the energy used to fit the data in figure S5 is incorrect.

The authors part:

The authors provide references to support the claim that the binding energy can indeed range from 17eV to 18eV.

3. Hf 4f will have two doublets due to spin-orbital splitting: one is 4f 5/2 which will have higher energy around 19.64 eV while 4f 7/2 which will have lower energy range from 16.7eV to 18.13eV as the author suggested. The assignment of higher energy peak for Hf 4f 7/2 peak is reasonable. It might be not fair to conclude that the fitting of figure S5 is flawed. It might underscore the discovery of this new Hf state during ALD deposition.

The referee part:

The inconsistency of peak assignment of 17.4eV to HfO₂. Where 16.7 eV should be HfO₂'s Binding energy instead of as the author's claim: the "extra In peak". And the referee believes that it is due to this flawed assignment that leads to the need for a new surface state of the Hf.

The authors part:

From the references (NIST Database), the Hf 4f peak indeed can exceed the 17eV limit. And the author also tries to fit the data with the constrained condition that provided by the referee (Using 16.7eV as the HfO₂ peak position), however, the outcomes that uses this boundary condition cannot match with the experimental data.

4. This paper presents new insight for the ALD process by using in-situ measurement with synchrotron source

The referee part:

Data fitting erroneous of the author, where peak positions of the fittings are placed too close to each other. And suggesting the author to monitor the signal of Hf 4d instead of Hf 4f, for that Hf 4d is more suitable for monitoring.

The authors part:

The condition in which the referee believes only held true for the lab X-ray source

5. The reported work (by Klejna and Elliot) mainly focus on the interaction between ligand exchange with oxide, and I quote from the same Klejna and Elliot paper:

“ The main emphasis here is on the interaction of the dimethylamido ligand (DMA) with the oxide surface with a view to finding potential redox properties of this ligand. “

In the paper, the authors focus on the adsorption of Hf-containing molecule on the surface before ligand exchange on the surface.

The referee part:

The mechanism proposed by the authors is flawed. With the computational paper from Klejna and Elliot, the referee believes that the dominated factor would be decomposition of ligand on the surface rather than the migration of the whole molecule.

The authors part:

It might not have the same circumstance as described in this report.

We are very grateful to reviewer 3 for his/her thorough evaluation of our work and of the discussion with reviewer 2. Reviewer 3 recommends acceptance of our manuscript and lists 5 important issues, where he/she totally supports our explanation of the observed findings and rejects the doubts mentioned previously by reviewer 2. We feel strongly encouraged in our research, even beyond the scope of this manuscript, by the detailed response of reviewer 3.